# Evaluation of the Immunoprotective Capacity of Five Vaccine Candidate Proteins against Avian Necrotic Enteritis and Impact on the Caecal Microbiota of Vaccinated Birds

**DOI:** 10.3390/ani13213323

**Published:** 2023-10-26

**Authors:** Sara Heidarpanah, Alexandre Thibodeau, Valeria R. Parreira, Sylvain Quessy, Mariela Segura, Marcelo Gottschalk, Annie Gaudreau, Tristan Juette, Marie-Lou Gaucher

**Affiliations:** 1Chaire de Recherche en Salubrité des Viandes (CRSV), Département de Pathologie et Microbiologie, Faculté de Médecine Vétérinaire, Université de Montréal, Saint-Hyacinthe, QC J2S 2M2, Canada; sara.heidarpanah@umontreal.ca (S.H.); alexandre.thibodeau@umontreal.ca (A.T.); sylvain.quessy@umontreal.ca (S.Q.); 2Swine and Poultry Infectious Diseases Research Centre (CRIPA), Faculté de Médecine Vétérinaire, Université de Montréal, Saint-Hyacinthe, QC J2S 2M2, Canada; mariela.segura@umontreal.ca (M.S.); marcelo.gottschalk@umontreal.ca (M.G.); annie.gaudreau.4@umontreal.ca (A.G.); 3Groupe de Recherche sur les Maladies Infectieuses en Production Animale (GREMIP), Faculté de Médecine Vétérinaire, Université de Montréal, Saint-Hyacinthe, QC J2S 2M2, Canada; 4Canadian Research Institute for Food Safety (CRIFS), Food Science Department, University of Guelph, Guelph, ON N1G 2W1, Canada; vparreir@uoguelph.ca; 5Faculté de Médecine Vétérinaire, Université de Montréal, Saint-Hyacinthe, QC J2S 2M2, Canada; tristan.juette@umontreal.ca

**Keywords:** *Clostridium perfringens*, necrotic enteritis, broiler chickens, surface-exposed antigenic proteins, immune response, gut microbiota

## Abstract

**Simple Summary:**

Necrotic enteritis is a complex gastrointestinal disease of broiler chickens that imposes a substantial economic burden on poultry producers worldwide. The causative agent is pathogenic strains of *Clostridium perfringens* type G. While numerous efforts to develop an effective vaccine against the disease were unsuccessful, a previous study by our group identified five potential vaccine candidate proteins. Herein, we assessed the relative contribution of the specific immunity raised by these proteins to protect broiler chickens against an experimental disease challenge and measured the impact on the intestinal microbiota. While it did not significantly impact the bacterial populations in the intestine, the protective capacity of the antibodies raised by the immunization of birds with the recombinant proteins should be further assessed using different necrotic enteritis induction models.

**Abstract:**

Avian necrotic enteritis is an enteric disease of broiler chickens caused by certain pathogenic strains of *Clostridium perfringens* in combination with predisposing factors. A vaccine offering complete protection against the disease has not yet been commercialized. In a previous study, we produced five recombinant proteins predicted to be surface-exposed and unique to necrotic enteritis-causing *C. perfringens* and the immunogenicity of these potential vaccine candidates was assessed in broiler chickens. In the current work, the relative contribution of the antibodies raised by these putative antigens to protect broiler chickens was evaluated using an experimental necrotic enteritis induction model. Additionally, the link between the immune response elicited and the gut microbiota profiles in immunized birds subjected to infection with virulent *C. perfringens* was studied. The ELISA results showed that the IgY antibody titers in vaccinated birds on days 21 and 33 were significantly higher than those on days 7 and 14 and those in birds receiving the adjuvant alone, while the relative contribution of the specific immunity attributed to these antibodies could not be precisely determined using this experimental necrotic enteritis induction model. In addition, 16S rRNA gene amplicon sequencing showed that immunization of birds with recombinant proteins had a low impact on the chicken caecal microbiota.

## 1. Introduction

*Clostridium perfringens* is recognized as both a zoonotic and animal pathogen, in addition to being largely accepted as a typical member of the intestinal microbiota represented by the large diversity of microorganisms living inside the gastrointestinal tract (GIT) of broiler chickens [1,2]. Recent advances in the use of high-throughput sequencing technologies have highlighted the pivotal role of this microbiota in the growth promotion of food-producing animals by facilitating the uptake of nutrients, stimulating the development of the immune system, and promoting an overall good health status, thus ensuring protection against pathogens [3]. Notwithstanding these protective attributes, the colonization of the intestine with virulent *C. perfringens* is a fundamental step in the pathogenesis of avian necrotic enteritis (NE), one of the most important gastrointestinal diseases of poultry worldwide [4].

Although several alternative control strategies—including probiotics, prebiotics, short- and medium-chain fatty acids, essential oils, organic acids, and plant extracts—were rapidly considered to prevent NE when the poultry industry started to rely less and less on the routine use of antibiotics, the search for a protective vaccine has always been an integral part of the research efforts for cost-effective control of the disease. While the initial focus for the development of an effective vaccine was on the *C. perfringens* alpha-toxin—a virulence factor regarded at that time as a key element in NE pathogenesis—it is now known, a little more than three decades later, that the establishment of the disease involves complex and tightly regulated interactions between *C. perfringens*, the chicken host, and the intestinal environment. Therefore, these components should be addressed as a whole [5,6,7,8,9,10,11,12]. To date, the immunogenicity and protective efficacy of several *C. perfringens* antigens—including alpha-toxin, NetB, TpeL, fructose 1,6-bisphosphate aldolase (FBA), pyruvate-ferredoxin oxidoreductase (PFOR), endo-beta-N-acetylglucosaminidase (Naglu), phosphoglyceromutase (Pgm), glyceraldehyde 3-phosphate dehydrogenase (GPD), and three predicted pilin structural subunits (CnaA, FimA, FimB)—have been evaluated. However, none of them have provided complete protection against an experimental NE challenge [13]. In addition, several studies have reported the effects of intestinal coccidial infection [14,15], diet types [15,16,17,18], probiotics [19,20], and different challenge models [21,22,23] on the gut microbiota of broiler chickens during experimental NE; however, none of them have examined the effects of vaccination against *C. perfringens* on the broiler chicken gut microbiota.

The most recent NE pathogenesis research findings have irrevocably contributed to the advancement of knowledge on the virulence of *C. perfringens*, in addition to leading the way in identifying potential vaccine targets, with a growing interest towards the bacterial features involved in the intestinal colonization by *C. perfringens*, an initial step of most bacterial diseases [24]. It is thus now known that the NE-causing competency of type G *C. perfringens* is linked to its ability to bind to the intestinal mucosa through surface-exposed structures. This feature appears as an initial critical prerequisite for the subsequent intervention of the Agr-like quorum-sensing system and the upregulation of the VirR/VirS regulon linked to the production of the NetB toxin, a virulence factor and antigen displayed as central in both NE pathogenesis and control through vaccination [6,25,26,27]. Consequently, other adherence factors, such as a collagen adhesin (CnaA) and fimbrial proteins (FimA and FimB), would then contribute to the deeper mucosal intestinal colonization by virulent *C. perfringens* [28]. 

With the aim of expediting the identification of *C. perfringens* appendages that could both be potentially involved in this first colonization step of the broiler chicken intestine during NE pathogenesis and that could be targeted to prevent the harmful effects of the pathogen on the intestine of birds, we previously used a comparative and subtractive reverse vaccinology approach. Using this approach, five new predicted putative antigenic surface-exposed proteins unique to NE-causing *C. perfringens* were shown to induce immunogenic responses in immunized birds [13]. The Western blotting and ELISA results also revealed that the raised antibodies were able to specifically recognize both the recombinant and native forms of the candidate proteins in pathogenic *C. perfringens*, respectively [13]. The current study proposes the evaluation of these predicted antigenic proteins by both documenting the protective ability of the antibodies raised following the exposition of the broiler chicken’s immune system to these candidate proteins and examining the impact of this immunity on the intestinal microbiota profiles of immunized birds when challenged in an experimental NE induction model. The objectives of the current work were then to (i) document the relative contribution of the specific immunity raised by these recombinant vaccine candidates using a combination of a vaccination trial and an experimental infection model for NE and (ii) study the gut microbiota profile of immunized birds submitted to *C. perfringens* experimental infection using Illumina MiSeq sequencing of the V4 region of the 16S rRNA gene.

## 2. Materials and Methods

### 2.1. Expression and Purification of Vaccine Candidate Proteins and Preparation of Whole-Cell Lysate from Virulent C. perfringens MLG_7820

All recombinant proteins except P509 were produced as previously described [13]. Briefly, amplification of the protein-encoding genes was performed through PCR and the resulting amplicons were sent for sequencing (McGill University and Génome Québec Innovation Centre, Montréal, QC, Canada). These same amplicons were cloned into the pET151/D-TOPO^®^ vector and transformed to *E. coli* TOP10 competent cells (Champion pET151 Directional TOPO^®^ Expression Kit, Invitrogen, Vacaville, CA, USA) according to the manufacturer’s specifications. The full-length form of P509 was predicted to contain one transmembrane helix between residues 12–34; thus, this region was excluded due to the obstacles related to purification. The gene encoding the truncated form of P509 was initially synthesized using an Oligo Synthesizer (Biolytic Lab Performance, Inc., Fremont, CA, USA) before being inserted into the pET-24a(+) vector using *Bam*HI and *Nde*I restriction enzymes (GenScript, Piscataway, NJ, USA). The recombinant plasmids were isolated using a QIAprep Spin Miniprep Kit (Qiagen, Hilden, Germany) and transformed to *E. coli* BL21 competent cells (Champion pET151 Directional TOPO^®^ Expression Kit, Invitrogen, CA, USA) for protein expression. Purification of the P153, P264, P509, and P537 recombinant proteins was conducted under native conditions and purification of P561 was performed under denaturing conditions, as described previously [13]. Preparation of whole-cell lysate from *C. perfringens* MLG_7820 and measurement of the protein concentration were carried out according to our previous work [13].

### 2.2. Bacterial Strains, Media, Culture Conditions, and Inoculum Preparation

The virulent strain *C. perfringens* MLG_7820 was grown anaerobically (Oxoid AnaeroGen gas packs (Thermo Fisher Scientific, Waltham, MA, USA)) on 5% sheep blood agar plates (Fisher Scientific, Ottawa, ON, Canada) for 18 h at 37 °C. Isolated colonies were inoculated into 100 mL of Cooked Meat medium (CMM; Oxoid Ltd., Basingstoke, UK (CM0081)), followed by incubation for 18 h at 37 °C under anaerobic condition. A 60 mL volume of the resulting culture was used to inoculate 2 L of Fluid Thioglycollate (FTG) broth (Biokar Diagnostics, Cedex, France), with subsequent anaerobic incubation for 18 h at 37 °C. *C. perfringens* cells were harvested through centrifugation at 3000× *g* for 20 min at room temperature (Thermo Scientific Sorvall Legend XTR with F14-6 × 250y Fixed-Angle Rotor), and the pellet was resuspended in FTG broth. One mL of this solution was kept for bacterial counting using serial dilutions (from 1 × 10^−1^ to 1 × 10^−7^) in 0.85% sterile saline. 

### 2.3. Birds Housing, Feeding, Experimental NE Disease Model, and Cecal Content Collection 

All experiments on chickens were approved by the Animal Ethics Committee of the Faculté de Médecine Vétérinaire of the Université de Montréal (certificate number: 20-Rech-2070) and carried out in accordance with the ARRIVE guidelines [29]. A total of 191 commercial day-old male Ross broiler chickens were purchased from a commercial hatchery (Couvoir Réal Côté, L’Ange-Gardien, QC, Canada) and were randomly divided into nine experimental groups (Table 1). All birds were fed a commercial diet containing 20% protein without antibiotics or anticoccidials (Meunerie Benjamin, Saint-Césaire, QC, Canada) and corresponding to the starter, grower, and finisher stages of a commercial broiler chicken diet for a period of 27 days. At 28 days of age, this diet was withdrawn for 12 h, before being replaced with a turkey diet containing 28% protein. The health status of the birds was monitored daily, and their body weight was also measured at the end of the trial. On day 29, 14 birds from each vaccinated group (groups 1 to 6) were randomly selected, individually weighed, and sacrificed according to the guidelines. The caeca were excised, and the content was collected in 2 mL cryovials (Sarstedt, Numbrecht, Germany) before being snap-frozen in liquid nitrogen and stored at −80 °C until DNA extraction for the purpose of a subsequent study. Starting from day 29, birds were orally inoculated once a day, for four consecutive days, with a 2 mL suspension of *C. perfringens* ranging between 10^8^ and 10^9^ colony-forming units (CFU)/mL depending on the inoculation day. All remaining birds were sacrificed at 33 days of age. In the same way as on day 29, the cecal content was harvested and the small intestine (from the duodenum to the ileum) was examined for gross NE lesions. Lesions were blindly scored from 0 to 5, according to Kulkarni et al. [30].

### 2.4. Vaccination of Broiler Chickens

All recombinant proteins to be injected were prepared in the morning of the inoculation; each protein was thawed on the ice and its concentration was quantified (Qubit Protein Assay kit (Invitrogen, Carlsbad, CA, USA) and Denovix QFX Fluorometer (Froggabio, Toronto, ON, Canada)) according to the manufacturer’s instructions. Broiler chickens were injected intramuscularly (pectoral muscle) on days 7, 14, and 21 with 200 μL of PBS containing 50 μg of the Quil-A adjuvant (InvivoGen, San Diego, CA, USA) and 50 μg of each recombinant protein. For the Mix group, every immunized bird received 50 μg of Quil-A adjuvant and 50 μg of each candidate protein (P153, P264, P509, P561), for a total of 200 μg of proteins. Birds from the MLG_7820 group were vaccinated with 200 μL of PBS containing 50 μg of proteins from the whole cell lysate preparation of *C. perfringens* MLG_7820 and Quil-A adjuvant (50 μg). Birds from the adjuvant control group received 50 μg of Quil-A in a 200 μL volume of PBS. Blood samples were collected in blood collection tubes (Covidien, Monoject blood collection tube, MA, USA) from all birds before each immunization and at day 33 (end of the trial). Blood tubes were kept at room temperature for 3 to 4 h before the serum was collected through centrifugation (3000 rpm, 19 °C, 10 min, Beckman Coulter with SX4750A Rotor (Beckman Coulter Inc, Miami, FL, USA)). Serum samples were frozen at −20 °C until ELISA analysis. 

### 2.5. DNA Extraction from Cecal Samples 

DNA extraction was conducted on 12 samples from each vaccinated group (samples from birds sacrificed post-challenge with *C. perfringens* MLG_7820 at 33 days old): 8 samples from the MLG_7820 group; 6 samples from the Quil-A group. The procedure was performed according to the protocol established by our laboratory [31]. Briefly, 200 mg of the thawed cecal samples along with 700 μL of lysis buffer [Tris-HCl 500 mM pH 8, EDTA 100 mM pH 8, NaCl 100 mM, SDS 1% (*w*/*v*)] were added to a 2 mL screw cap tube (MP Biomedical, Solon, OH, USA) containing 500 mg of 0.1 mm silica spheres. All samples and a negative control tube (containing only 900 μL of lysis buffer) were submitted to mechanical lysis using a FastPrep-24™ 5G Instrument (MP Biomedicals, VWR, Ville Mont-Royal, QC, Canada) for three runs of 60 s each, at 6 m/s, with a 5 min pause between each run. Samples were kept on ice during intervals. A subsequent thermal lysis (20 min at 95 °C, Dry Bath Incubator, Fisher Scientific) was also performed. The supernatant was collected after centrifugation at 14,000× *g*, 4 °C for 15 min and a standard phenol/chloroform method was used to complete DNA extraction. The quantity and quality of the extracted DNA were measured using the QFX Fluorometer (Froggabio, Toronto, ON, Canada) and Nanodrop 1000 device (Fisher Scientific, Ottawa, ON, Canada), respectively, according to the manufacturer’s instructions. DNA samples were kept at −80 °C until analysis through 16S rRNA gene amplification. 

### 2.6. 16S rRNA Gene Amplification, Amplicon Sequencing, and Bioinformatics Analysis 

Amplification of a 292 bp fragment from the V4 region of the bacterial 16S rRNA was conducted using 6 μL of 5× SuperFiTM GC Enhancer (Fisher Scientific, Ottawa, ON, Canada), 6 μL of 5× SuperFiTM Buffer (Fisher Scientific, Ottawa, ON, Canada), 0.6 μL of 10 mM dNTP mix (Fisher Scientific, Ottawa, ON, Canada), 1.8 μL of 10 μM forward primer (5′-ACACTGA CGACATGGTTCTACAGTGCCAGCMGCCGCGGTAA-3′), 1.8 μL of 10 μM reverse primer (5′-TACGGTAGCAGAGACTTGGTCTGGACTACHVGGGTWTCTAAT-3′) (Invitrogen, CA, USA), 0.6 μL of 20 mg/mL PierceTM bovine serum albumin (Fisher Scientific, Ottawa, ON, Canada), 0.3 μL of 2 U/μL Platinum SuperFi DNA Polymerase (Fisher Scientific, Ottawa, ON, Canada), 2.5 μL of DNA (12.5 ng), and ddH_2_O in a total reaction volume of 30 μL. Eight negative controls (one for each DNA extraction that was conducted on different days) to confirm the quality of the DNA extraction procedure; two negative controls (sterile water), and two positive controls (ZymoBIOMICS Microbial Community DNA Standard (Cedarlane, Burlington, ON, Canada)) were also included for the PCR approach. Cycling conditions were as follows: an initial denaturation step at 95 °C for 5 min, 23 cycles of denaturation at 95 °C for 30 s, annealing at 55 °C for 30 s, and elongation at 72 °C for 3 min. The reaction was completed through an extension step at 72 °C for 10 min. A volume of 4 μL of each PCR product was subjected to electrophoresis using 1% agarose gels containing 0.01% SYBR Safe DNA gel stain (Fisher Scientific, Ottawa, ON, Canada). Amplicons were sent to the Génome Québec Innovation Centre (Montreal, QC) for DNA sequencing of the V4 region using an Illumina MiSeq PE250 (2 × 250 bp V3 chemistry) platform (Illumina, San Diego, CA, USA) [31]. 

The obtained raw sequences were processed using the MOTHUR standard operating procedure (v. 1.14.3) [32], as previously described [33]. Briefly, forward and reverse reads were merged into contigs for every sample and ambiguous or too-long reads were eliminated. Potential chimeras were discarded using VSEARCH [34]. Unique sequences were aligned to the SILVA database version 132 (Mothur-formatted). The remaining reads were clustered into operational taxonomic units (OTU) at 97% sequence similarity and the obtained OTUs were classified using the Mothur-formatted Ribosome Database Project (RDP) trainset version 18. Finally, classified sequences as Eukaryota, Archaea, chloroplasts, mitochondria, and unclassified sequences were eliminated [31].

### 2.7. Measurement of Serum Antibody Levels by ELISA 

Antibody titers in chicken sera representing both the specific (candidate proteins) and non-specific (adjuvant only) immunity were determined using an indirect ELISA approach. Briefly, the wells of Maxisorp 96 well flat-bottom Immuno plates (Thermo Scientific, NY, USA) were coated with 100 µL of purified recombinant protein (0.5 μg/well) diluted in 50 mM carbonate/bicarbonate coating buffer (pH 9.6). The plates were then incubated for 1 h at 37 °C, followed by incubation at 4 °C overnight. After three washes with washing buffer (PBS containing 0.05% Tween 20), the plates were blocked with 100 µL of blocking buffer (washing buffer containing 0.3% casein; Sigma-Aldrich, USA), and then incubated for 2 h at 37 °C. A volume of 100 µL of 2-fold serially diluted chicken sera (ranging between 1:50 and 1: 3,276,800) in blocking buffer was added to each well after being washed three times with the same washing buffer. The plates were then incubated at 37 °C for 2 h, followed by washing 3 times with washing buffer. Subsequently, a volume of 100 μL of goat anti-chicken IgY horseradish peroxidase (HRP)-conjugated polyclonal antibody (Bethyl Laboratories, Montgomery, TX, USA, cat. no A30-104P; diluted 1:8000 in wash buffer) was applied to each well. Following 1 h of incubation at 37 °C and 3 washes with wash buffer, 100 μL of 3,3′,5,5′-tetramethylbenzidine (TMB; Life Technologies, Inc., Carlsbad, CA, USA) substrate was added to each well, and the plates were incubated for 15 min at room temperature in the dark. The reaction was stopped with 100 μL/well of 0.18 M H_2_SO_4_, and the absorbance at 450 nm was read using a spectrophotometer (EZ Read 400, Biochrom, Cambridge, UK), according to the manufacturer’s instructions. 

### 2.8. Statistical Analysis

A post-hoc test with Benjamini-Hochberg correction was used for statistical analysis of the ELISA results obtained from serum samples collected at 7, 14, 21, and 33 days of age. The body weight of the birds was statistically analyzed using a mixed linear model with the body mass of individuals as the dependent variable and the group as the explanatory variable. The identity of the individuals was indicated as a random factor. A post-hoc test of multiple comparisons between the groups with a Benjamini-Hochberg correction on the *p*-values was subsequently applied. Necrotic enteritis lesion scores were statistically analyzed using the non-parametric Kruskal-Wallis test before a post-hoc test of multiple comparisons between the groups with a Benjamini-Hochberg correction was carried out.

Alpha and beta diversity analyses were performed to assess the impact of the treatments on the bird’s microbiota at day 33 using rarefied data in RStudio v.1.4.1103 (RStudio, Inc., Boston, MA, USA), according to the standard operating procedure (SOP) used in a previous study [35]. Three alpha diversity indices (observed OTUs, Shannon, and inverse Simpson) were measured using the estimate_richness function of the phyloseq package in R [36], followed by a Kruskal-Wallis test with multiple comparison. Beta-diversity was calculated using the Bray-Curtis (based on the relative abundance of each genus distance metric and then plotted on a two-dimensional map with non-metric multidimensional scaling (NMDS)). Statistically significant differences between groups were tested using the permutational multivariate analysis of variance (PERMANOVA) via the pairwise ADONIS function within the R package Vegan [37]. Biomarkers were also analyzed using MaAsLin2 [38], using NEGBIN as an analysis method on unrarefied data, and all *p*-values were adjusted for multiple comparisons using the false-discovery rate (FDR) or *q*-value method, with *q* lower than 0.20 and *p* value lower than 0.05 considered significant.

## 3. Results

### 3.1. Expression and Purification of Vaccine Candidate Proteins and Preparation of Whole-Cell Lysate of Virulent C. perfringens MLG_7820

Various concentrations of purified proteins were produced for the in vivo assay: 2.5 μg/μL for P537, and 3 μg/μL and 2 μg/μL for P153 and P264, respectively (Figure 1). The protein concentration of the whole-cell lysate derived from *C. perfringens* MLG_7820 was 14.6 μg/μL. Table 2 shows the name, size, cellular localization, and VaxiJen score of the candidate proteins used for the in vivo assay.

### 3.2. Bacterial Strains, Media, Culture Conditions, and Inoculum Preparation

The birds received a concentration of *C. perfringens* varying between 1 × 10^8^ and 2.2 × 10^9^ CFU/mL: 5.6 × 10^8^ CFU/mL and 1.4 × 10^9^ CFU/mL for the first and second days of inoculation, respectively, and 1 × 10^8^ CFU/mL and 2.2 × 10^9^ CFU/mL for the third and fourth inoculations, respectively.

### 3.3. NE Lesion Scoring and Average Body Weight Gain of Broiler Chickens

No mortality or clinical signs associated with NE, such as depression, diarrhea, or ruffled feathers, was observed before or after the challenge. Table 3 presents the number of birds per group, the NE-associated gross lesion scores observed at 33 days of age in the different groups, the average lesion score, and the average body weight for each group. Among the 103 birds examined for establishing a macroscopic lesion score, 71 showed no lesion, while 16, 14, 1, and 1 bird revealed lesion scores of 1, 2, 3, and 4, respectively (Figure 2). The mean lesion score observed in the birds from the MLG_7820 group was 1.22, which was the highest among the experimental groups. In contrast, the lowest average lesion scores were attributed to the birds from the Bacitracin (0), P561 (0.07), and Quil-A groups (0.17). 

Regarding the final body weight (Table 3), the Bacitracin group was identified as the heaviest, with a mean body weight of 2790 g, followed by the P537 group (2087 g). Conversely, the lowest average body weight was recorded from the P561 group (1900 g), followed by the Quil-A group (1925 g). 

Table 4 shows the results of multiple comparisons between the groups for the NE lesion scores. The post-hoc test followed by Benjamini–Hochberg correction indicates a statistically significant difference (*p* < 0.05) between the MLG_7820 and P561 groups; the birds from the MLG_7820 group exhibited more severe lesions. A significant trend (0.05 ≤ *p* < 0.10) was also observed between the MLG_7820 and Bacitracin groups and between the P561 and P537 groups, where the birds from MLG_7820 and P537 had higher lesion scores, respectively.

Table 5 indicates the results of multiple comparisons between the groups for body weight. According to the model, there is an effect of the group on the body weight of the birds (LMM: Χ2 = 177.80, Df = 10, *p*-value < 0.001). A statistically significant difference in the body weight between the birds from the Bacitracin group and all other groups was found (*p*-value < 0.001), with the former being heavier. A significant difference (*p*-value < 0.05) was also observed when comparing the body weight of the birds in the P561 and P537 groups, with the latter being heavier (187 g). 

### 3.4. Measurement of Serum Antibody Levels by ELISA

The statistical analysis of the ELISA results revealed significant differences between the IgY antibody titers on days 21 and 33 in comparison to days 7 and 14 for the birds immunized with P537 (Figure 3). For the birds injected with P264, the antibody levels at 14, 21, and 33 days of age were significantly higher than those found at day 7. A third immunization at day 21 significantly increased the IgY antibody levels at day 33 for this group. The highest mean IgY titers were observed in the birds vaccinated with P537 (1.31 × 10^5^ and 2.52 × 10^5^) and P264 (7.86 × 10^4^ and 3.76 × 10^5^) on days 21 and 33, respectively. Conversely, the lowest mean IgY antibody levels were found in the broiler chickens vaccinated with P153 (1.34 × 10^2^ and 6.74 × 10^2^), P509 (6.10 × 10 and 6.02 × 10^2^), and P561 (1.57 × 10^2^ and 1.13 × 10^3^) on days 21 and 33, respectively. Significant changes were noted in the IgY titers between day 33 and days 7, 14, and 21 for the birds from the P509 group. Similar results were also documented in the birds immunized with P561, and significant changes between day 33 and days 7, 14, and 21 were also seen for this group. For the birds from group 153, a significant difference between days 14 and 33 was reported, whereas no significant change between days 7 and 33 was observed. This trend was also noted during the first immunization trial [13]. The IgY antibody response of the birds receiving the adjuvant alone was negligible at 33 days of age. Sera collected from the birds immunized with the mixture of proteins were tested separately against each protein. Again, the highest antibody response was obtained in the P264 and P561 group, and the lowest antibody level was achieved in the P153 and P509 groups at 33 days of age. Significant differences between the antibody titers at day 33 and day 7 were noted for all candidates (Figure 3).

### 3.5. 16S rRNA Gene Amplicon Metagenomic Sequencing

A total of 4,409,736 sequences were retained after cleaning. The average number of reads was 47,416.52 per sample and the highest and lowest numbers were 68,047 and 33,327, respectively. The composition of the positive controls was as expected based on the composition provided by the manufacturer (see Appendix A). When all of the samples are compared together in an NMDS plot (see Appendix A), with the exception of one negative control that denotes some contamination, all of the controls were grouped apart from the study samples. 

A total of four phyla were identified in the samples with more than 1% abundance. Among all of them, *Ruminococcaceae*, *Lachnospiraceae*, and *Bacteroidaceae* had the highest relative abundance of identified families of the chicken caecal contents analyzed in all of the experimental groups (Figure 4). *Ruminococcaceae* was not statistically different among the experimental groups (FDR adjusted *p*-value = 0.22, Figure 5A), while the *Bacteroidaceae* family was significantly higher (FDR adjusted *p*-value < 0.05) in the birds immunized with P153 and P537 when compared to the Quil-A control group (Figure 5B,C). *Lachnospiraceae*, *Erysipelatoclostridiaceae*, and *Oscillospiraceae* were lower in the MLG_7820 group compared to all of the other experimental groups, although this was not statistically significant (FDR adjusted *p*-value > 0.05, Figure 5D–F). A meaningful increase in members of *RF39* and *Clostridia* (unclassified) in the P561 experimental group was observed in comparison to the Quil-A group (FDR adjusted *p*-value < 0.01, Appendix A, respectively). Conversely, a significant decrease in Enterobacteriaceae in the P561 experimental group was found when compared to the Quil-A control group (FDR adjusted *p*-value < 0.01, Appendix A). A considerable increase in the population of the *Lactobacillaceae* was seen in the P537 group compared to the Quil-A control group (FDR adjusted *p*-value < 0.01, Appendix A). *Clostridiaceae* were found to be higher in the P537 and MLG_7820 groups, although this was not statistically significant (FDR adjusted *p*-value > 0.05, Appendix A). In addition, *Erysipelotrichaceae* and *Clostridia UCG-014* were different across the groups, but not significantly so (FDR-adjusted *p*-value > 0.05, Appendix A).

### 3.6. Alpha and Beta Diversities

The alpha diversity analysis (Appendix A) based on the observed OTUs showed no significant (*p*-value > 0.05) differences among the experimental groups (Table 6). However, the Shannon index was statistically different between several groups (*p*-value < 0.05 for P509 vs. MLG_7820, P509 vs. P153, P509 vs. Mix, P509 vs. P537, P509 vs. P561) (Table 7). The samples from the P509 group showed lower Shannon diversity (lower bacterial population evenness) than the P537, P561, and Mix groups, but higher Shannon diversity compared to the 153 and MLG_7820 groups (Table 8). 

Likewise, the inverse Simpson diversity index was significantly different across several groups (*p*-value < 0.01 for MLG_7820 vs. Mix and *p*-value < 0.05 for MLG_7820 vs. P561) (Table 9). According to Table 8, the samples from the MLG_7820 group exhibited lower inverse Simpson diversity (lower bacterial richness and evenness) compared to the Mix and P561 samples (Table 8). 

The beta diversity was affected by the experimental treatments and is graphically represented using a 2D NMDS (Figure 6). Table 10 shows the ADONIS analysis between different groups. 

## 4. Discussion

NE is a severe enteric disease in broilers that causes a significant increase in flock mortality and can result in mucosal damage and a microbial shift in the small intestine, leading to decreased zootechnical performances. It thereby has an unfavorable impact on birds’ welfare and profitability for poultry producers [39,40,41]. Although pathogenic *C. perfringens* strains are the main causative agent of NE, the presence of other predisposing factors is also crucial to experimentally inducing the disease. Avian coccidiosis, an enteric parasitic disease caused by one or more *Eimeria* species (e.g., *E. acervulina*, *E. maxima*), is one of the best-known predisposing factors for NE [42]. Vaccination is a promising non-antibiotic strategy as a preventive measure, but despite extensive efforts over recent years, no commercial vaccine for the control of NE has been developed to date [42,43]. 

In a previous study conducted by our group [13], the recombinant form of five proteins predicted to be surface-exposed and unique to NE-causing *C. perfringens* was evaluated for its immunogenicity in broiler chickens. In this work, the same candidate proteins, except for P509, were evaluated for their relative contribution to the protection of broiler chickens and their impact on the caecal microbiota of these birds using an experimental NE-induction model. For P509, a truncated form excluding a hydrophobic transmembrane domain, was used to facilitate purification. Similar difficulties with the cloning, expression, and purification of the full-length coding sequence of pilin subunits were also reported by other authors; this is probably due to the existence of the hydrophobic N-terminus [44,45,46]. It has also been reported that the deletion of this hydrophobic N-terminus can lead to the overexpression of the protein product, which can in turn be recovered under its soluble form [44,45,46]. For P509 and P561, which are both identified as prepilin N-terminal cleavage methylation domain proteins, the presence of a hydrophobic N-terminus was hypothesized to be the main interfering factor that might have contributed to the obstacles encountered in the expression and purification of the recombinant form of these candidate proteins [13]. 

The results from our first study indicated that immunization of the birds with P537, P509, and P264 induced the highest IgY titers at 35 days of age [13]. Conversely, P153 and P561 evoked the lowest antibody responses in the immunized broiler chickens at the same age [13]. 

The ELISA results obtained during the current study confirmed the trends observed for the IgY titers measured during the previous immunization trial. The highest antibody response was achieved at 33 days of age through immunization with P537 [13]. The immunization of broiler chickens with the full-length form of P509 was correlated with the second mean highest IgY titers (2.41 × 10^5^) at the end of the first immunization trial. After the removal of its transmembrane region, the IgY response elicited by this candidate protein reached a mean value of 6.02 × 10^2^ in the birds analyzed at 33 days of age. This result was unexpected given the immunogenic score predicted by the VaxiJen software v 2.0, which increased from 0.86 to 0.91, when comparing the full-length and the truncated sequence of P509. Thus, the lowest IgY titers obtained during this second immunization trial were measured in the birds immunized with P509. The conformational stability of a protein is known to act as an important factor in the stimulation of the immune system. In addition, despite reports that the removal of 28 residues or more from the hydrophobic first half of the N-terminal coding sequence of pilin proteins in order to improve their solubility had minimal impact on the protein structure itself, some authors have also shown that modification of the N-terminal sequence of pilin proteins was modifying their solubility, their role in the pilus assembly, and thus, potentially, their ability to induce an immune response [47,48,49]. The comparatively weak fold change in the IgY titers measured from the birds immunized with a truncated form of P561 at 7 and 33 days of age, another pilin-associated protein, during both immunization trials further supports this hypothesis [13]. The difficulties encountered by our group in obtaining a pure product of the recombinant form of P509 during the first immunization trial might have contributed to the magnitude of the response generated in the immunized birds through the exposition of their immune system to some other bacterial components not being removed in the purification step. This could also explain the variations in the mean values and the fold change in IgY titers observed between both trials [13]. 

The IgY titer values clustering for each time point and for each of the vaccination groups leads us to think that the antigen delivery was consistent, and the similar IgY response patterns observed for both immunization trials strengthen this affirmation [13]. Previous vaccination studies assessing the protective ability of other *C. perfringens* recombinant proteins stressed the fact that a strong serum IgY response does not always positively correlate with a protective mucosal immunity through the action of secretory IgA. We could presume that the low average lesion score observed for all the immunized groups in the current study could support such an extrapolation [30]. When considering an average lesion score for birds receiving the Quil-A adjuvant only, which should range between 1 and 4,depending on the severity of the challenge, as well as the variability in the lesion scores among the birds from different groups, we could hypothesize that the NE-induction model used might compare to a field challenge, during which some broiler chickens will reveal severe intestinal lesions, sometimes resulting into death, while some other birds will not show any particular impairment at both the clinical and histological levels [30,50,51,52,53,54]. Thus, while broiler chickens submitted to a severe experimental NE challenge might benefit from some specific immunity for fighting the disease, the induction model used in the current study made the distinction between the contribution of the non-specific (adjuvant only) and specific immunity (candidate proteins) less clear. Different experimental models can be used for the induction of NE in broilers, most of them relying on a combination of predisposing factors and inoculations with a virulent strain of *C. perfringens* over a period of 1 to 5 days [55]. While the current study used an oral gavage with a suspension of bacterial pellets of the virulent strain MLG_7820 once daily for four consecutive days, other authors achieved higher lesion scores using whole bacterial broth cultures mixed with the feed served twice daily for infecting the birds. This approach most likely increases the bird’s exposure to the preformed toxins and secreted proteins of *C. perfringens* and might also contribute to the virulence of the pathogen and the severity of the lesions observed [53,56,57]. 

As the aim of the current study was to document the relative contribution of the specific immunity raised by the studied recombinant proteins to prevent NE, with no other interference with the immune system, a high animal-based protein diet was used as a predisposing factor in contrast to other studies using an *Eimeria* infection as the predisposing factor [58]. It should be pointed out that the birds with the highest average lesion score belonged to the MLG_7820 group, highlighting the fact that the exposition of the birds’ immune system to the whole diversity of *C. perfringens* proteins and the stimulation of the production of a plethora of antibodies does not necessarily translate into protection against *C. perfringens*. 

Whereas only birds from the bacitracin group revealed a significantly higher body weight when compared to the other experimental groups, we can safely hypothesize that the injection of the recombinant protein followed by the stimulation of an immune response did not significantly impact the growth performance of the birds—a critical feature of any vaccine strategy for which the intended purpose is a commercial application. It is well established that antibiotics such as bacitracin methylene disalicylate (BMD) promote weight gain and feed efficiency in broiler chickens by increasing the villus height throughout the small intestine, ameliorating the digestion of dietary components, and modulating the intestinal microbiota [59,60,61]. 

Intestinal health is directly linked to the composition of the microbiota that interacts with the host [62]. It has been demonstrated that the broiler chicken gut microbiota plays central roles in providing the host with nutrients and vitamins (e.g., vitamin K and biotin), improving intestinal integrity, modulating immunity, and preventing enteric pathogens from colonizing the GIT [62]. A high proportion of *Ruminococcaceae* and *Lachnospiraceae*, which belong to the Firmicutes phylum, was identified in the microbiota of the birds from all of the groups in this trial. The *Ruminococcaceae* family, which contains a range of highly oxygen-sensitive butyrate-producing species [63], was the most prevalent family identified in the cecum of the birds across all the experimental groups in this study. This family is associated with the production of short-chain fatty acids (SCFA) such as butyrate and the degradation of plant materials; hence, it is correlated with better feed conversion and favorable productivity outcomes [63]. In addition, *Ruminococcaceae* were found to be highly abundant in the cecal microbiota of chickens with a low feed conversion ratio (FCR) [63]. Likewise, *Lachnospiraceae*, the second most dominant family identified in all the experimental groups, is also involved in butyric acid production [63]. The *Oscillospiraceae* and *Erysipelatoclostridiaceae* families, which belong to the Firmicutes phylum and also contain butyrate-producing members, were identified in all of the experimental groups. [64]. There is accumulating evidence indicating that butyrate has an anti-inflammatory property, stabilizes intestinal integrity, and strengthens the epithelial barrier [65]. Additionally, it is a pivotal signaling molecule in the GIT and serves as an energy source for epithelial cells [66]. It has also been reported that butyrate decreases the incidence and severity of NE when used as a feed additive; thus, the high abundance of butyrate-producing bacteria in the microbiota of birds involved in the current trial could potentially contribute to the reduced severity of the observed lesion scores [65]. The immunization of the birds with recombinant proteins did not considerably affect the abundance of *Ruminococcaceae*, *Lachnospiraceae*, *Oscillospiraceae*, and *Erysipelatoclostridiaceae* compared to the Quil-A control group (FDR adjusted *p*-value > 0.05) (Figure 5A,D–F). In contrast, a reduction in the relative population of *Lachnospiraceae*, *Erysipelatoclostridiaceae*, and *Oscillospiraceae* was observed in the MLG_7820 group, which was the group with the highest lesion score (Figure 5D–F). Thus, the existence of a link between the high NE lesion score in the MLG_7820 group and the reduction in beneficial butyrate-producing bacteria in the same group could be considered. Consistent with this observation, Wu et al. demonstrated that NE predisposing factors such as a high-protein fishmeal diet and *Eimeria* infection reduced the abundance of butyrate-producing *Ruminococcaceae* and *Lachnospiraceae* families of bacteria in chickens [15]. 

The P537 group, which was characterized by the highest antibody titers, showed two significant changes compared to the control group. The populations of *Lactobacillaceae* and *Bacteroidaceae* increased noticeably in the birds receiving P537 (FDR adjusted *p*-value < 0.05). Also, the birds from this group revealed the second highest mean body weight and the second highest mean lesion score (0.85) among the experimental groups. The superior abundance of *Lactobacillaceae* and *Bacteroidaceae* in the P537 group might be a direct impact of the experimental NE challenge. This observation was also made by Lacey et al., who reported both an increase in the proportion of *Lactobacillus* bacteria in broiler chickens presenting NE lesions and a negative correlation between the abundance of this bacterial population and the feed conversion ratio of birds [41]. *Lactobacillaceae* is a diverse family of lactic acid-producing bacteria found within the gut of both vertebrates and invertebrates [67]. 

Conversely, the Bacteroidota phylum, represented by members of the *Bacteroidaceae* family, can reduce nutritional absorption and lead to dysbiosis [68,69]. Xu et al. showed that the population of Bacteroidota and Proteobacteria phyla increased with the severity of NE [70]. Hence, it could be supposed that the higher abundance of this family correlates with a greater susceptibility to developing higher NE lesions (lesion score of 0.85) for birds belonging to this group [41]. 

One significant change observed in the birds from the P561 group (the group with the lowest average lesion score) was a reduction in the abundance of *Enterobacteriaceae*. It could be assumed that the low abundance of the *Enterobacteriaceae* family correlates with the low lesion score observed in this group, as Bortoluzzi et al. revealed that a NE challenge increased the frequency (*p* = 0.01) of *Enterobacteriaceae* family members [71,72]. Organic acids and essential oils have been shown to improve performance and digestive function in broilers under NE and to reduce the abundance of some harmful families, such as *Enterobacteriaceae*; these findings further supporting this hypothesis [72].

## 5. Conclusions

Taken together, the results obtained herein are promising and suggest that non-specific immunity might contribute to a certain level of protection when broiler chickens are exposed to a milder NE challenge, adhering more closely to the exposition of birds to virulent *C. perfringens* in field conditions. Although the role of the specific immunity attributed to the antibodies raised through immunization with the recombinant proteins could not be precisely determined using this mild experimental NE induction model, additional trials with increasing severity levels should be repeated to bring to light the coverage offered by the specific immunity attributed to the tested vaccine candidate proteins. In addition, other vaccine delivery routes, including maternal immunization of layer hens, oral immunization, and *in ovo* vaccination, should be considered to cater for the needs of the poultry industry while ensuring a low impact on the chicken caecal microbiota is maintained.

## 6. Patents

The vaccine candidates mentioned in this article are the subject of a US Patent Application, 63/486,749, which was filed on 24 February 2023, and is entitled “RECOMBINANT VACCINE PROTEINS FOR THE PREVENTION OF AVIAN NECROTIC ENTERITIS”.

## Figures and Tables

**Figure 1 animals-13-03323-f001:**
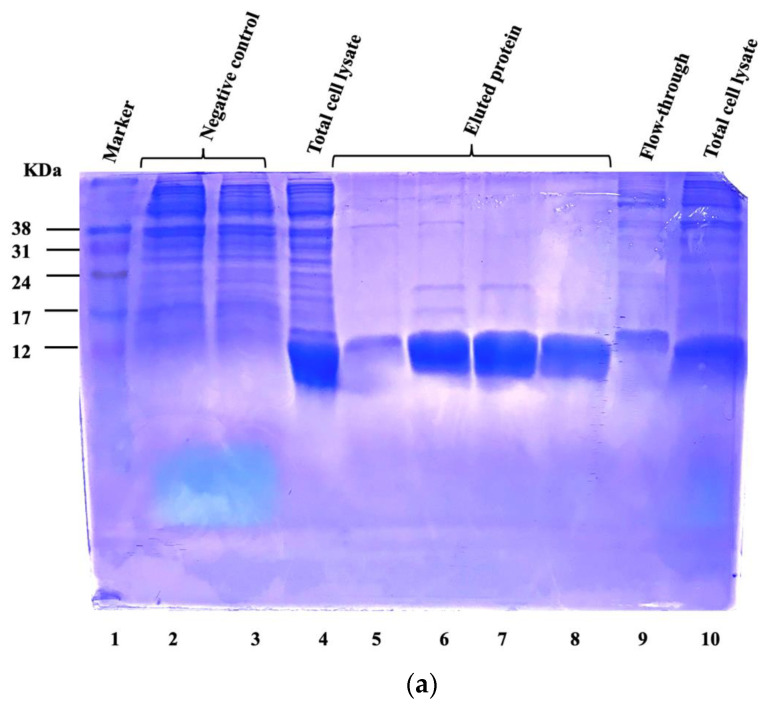
Coomassie blue stained SDS-PAGE gels showing purification of His-tagged recombinant proteins by Ni-NTA affinity chromatography. A volume of 15 µL of crude *E. coli* extract and fractions of (**a**) P153, (**b**) P264, (**c**) P509, (**d**) P537, and (**e**) P561 were loaded onto 15% SDS-polyacrylamide gels. The size of the protein of interest is 4 KDa higher than the predicted size due to the insertion of V5 and His-tag at the N-terminal of each coding sequence. Negative control sample in Figure 1a corresponds to the crude cell lysate from BL21 cells that were transformed with pET empty vector. The uncropped original gels are presented in Appendix A.

**Figure 2 animals-13-03323-f002:**
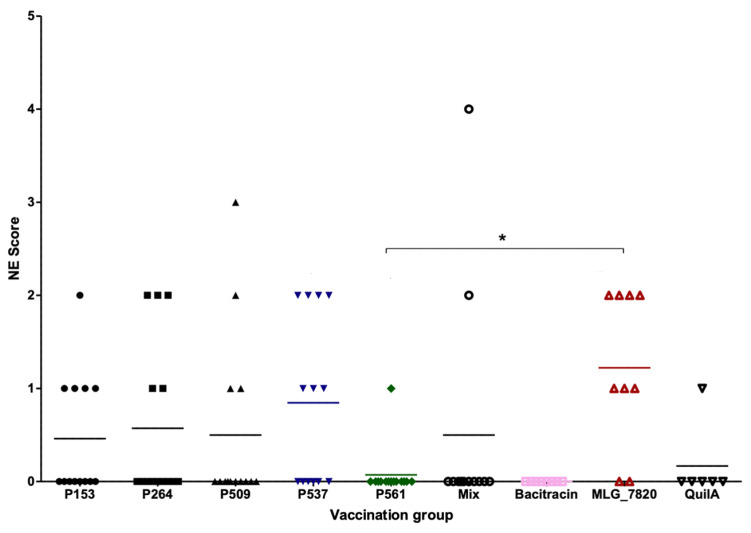
Lesion scores in birds submitted to NE-experimental infection and immunized with adjuvant alone, recombinant form of the candidate proteins, and whole-cell lysate from virulent *C. perfringens* MLG_7820. The bacitracin group was not immunized before the challenge (* *p* < 0.05 and lines correspond to the average lesion score).

**Figure 3 animals-13-03323-f003:**
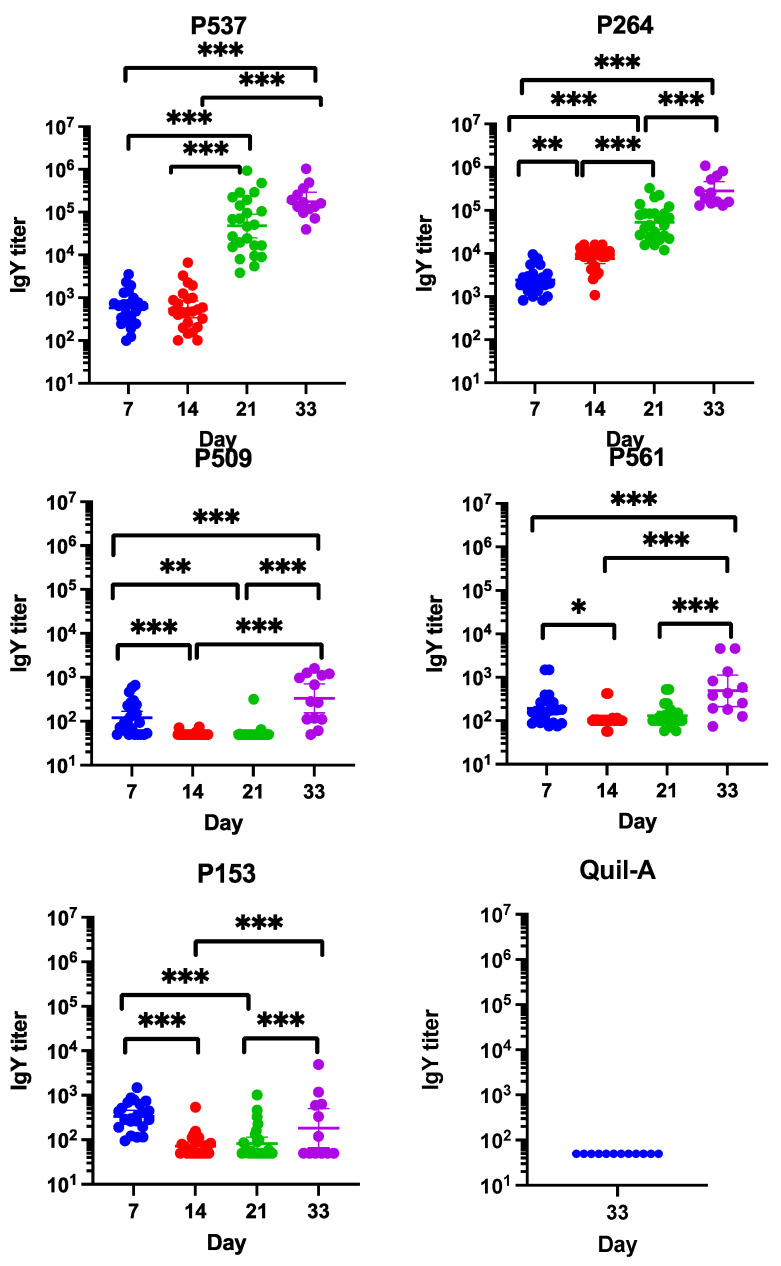
ELISA antibody responses in birds immunized with indicated recombinant proteins mixed with the adjuvant Quil-A. The negative control group (Quil-A) was injected with the adjuvant alone. Birds in Mix group received 50 μg of each of the four candidate proteins (P153, P264, P509, P561, for a total of 200 μg of proteins) and 50 μg of Quil-A adjuvant. Immuno plate MaxiSorp wells were coated with 0.5 μg of each recombinant protein. The sera obtained from birds from the Mix group were tested four times separately against each recombinant protein. The number of asterisks indicates the level of significance: * *p* ≤ 0.05; ** *p* ≤ 0.01; *** *p* ≤ 0.001.

**Figure 4 animals-13-03323-f004:**
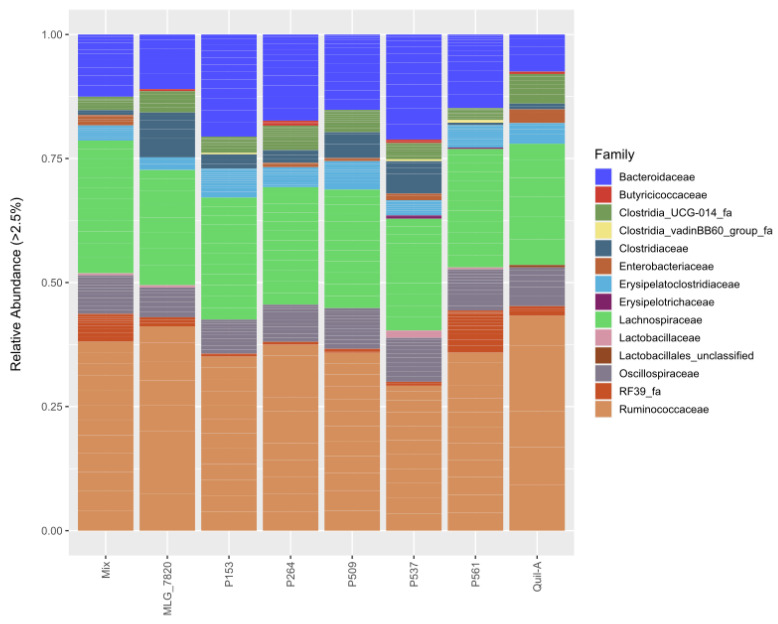
Relative abundance of the major bacterial genera identified in the ceca of treated birds and submitted to NE infection. Only bacterial genera representing at least 2.5% of the total reads are shown.

**Figure 5 animals-13-03323-f005:**
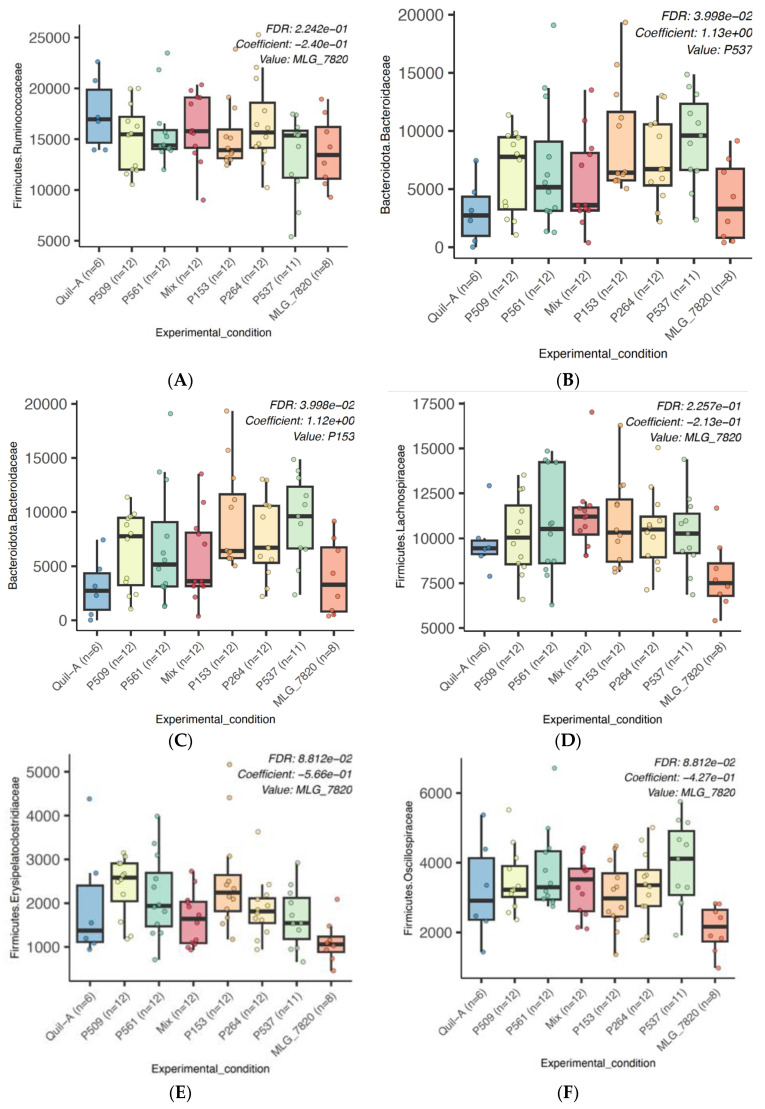
Statistically significant changes (*p*-value < 0.05) in key bacterial families of interest across experimental groups.

**Figure 6 animals-13-03323-f006:**
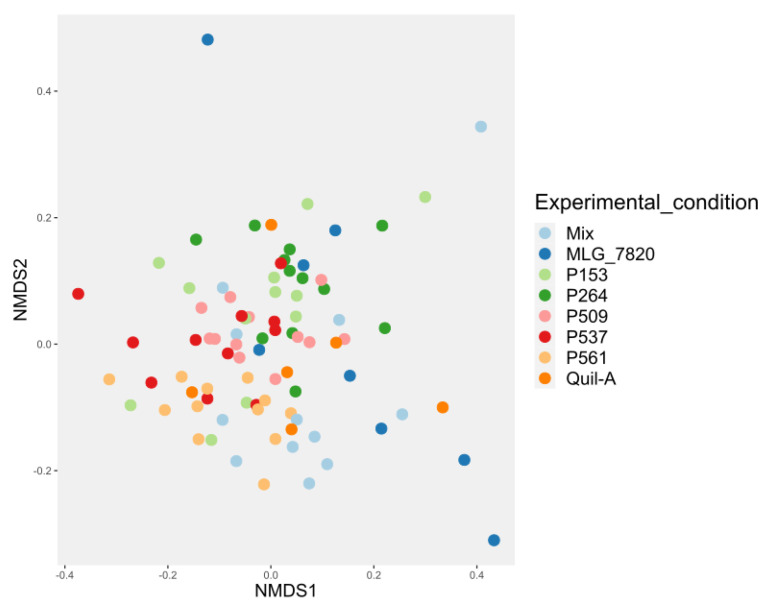
Non-metric multidimensional scaling plot (NMDS) illustrating the dissimilarity between sample types using Bray–Curtis index.

**Table 1 animals-13-03323-t001:** Name, number of birds per group, days of immunization, dose of vaccine, and days of serum collection.

Groups	No. of Birds per Group	Days of Vaccination	Dose of Vaccine	Days of Serum Collection
P153	28	7, 14, 21	Quil-A (50 μg) + P153 (50 μg)	7, 14, 21, 33
2. P264	28	7, 14, 21	Quil-A (50 μg) + P264 (50 μg)	7, 14, 21, 33
3. P509	28	7, 14, 21	Quil-A (50 μg) + P509 (50 μg)	7, 14, 21, 33
4. P537	28	7, 14, 21	Quil-A (50 μg) + P537 (50 μg)	7, 14, 21, 33
5. P561	28	7, 14, 21	Quil-A (50 μg) + P561 (50 μg)	7, 14, 21, 33
6. Mix	28	7, 14, 21	Quil-A (50 μg) + P153 (50 μg) + P264 (50 μg) + P509 (50 μg) + P561 (50 μg)	7, 14, 21, 33
7. Quil-A	6	7, 14, 21	Quil-A (50 μg)	7, 14, 21, 33
8. Bacitracin	8	-	-	33
9. MLG_7820	9	7, 14, 21	Quil-A (50 μg) + whole-cell lysate of *C. perfringens* MLG_7820 (50 μg)	7, 14, 21, 33

**Table 2 animals-13-03323-t002:** Name, size, cellular localization, and VaxiJen score of candidate proteins that were used in in vivo assay.

Name	Size (KDa)	Location	VaxiJen Score
P153	5.6	Extracellular/Cytoplasmic membrane	0.69
P264	9.2	Extracellular/Cytoplasmic	1.47
P509	13	Extracellular	0.91
P537	21	Cytoplasmic	0.21
P561	17	Extracellular	0.64

**Table 3 animals-13-03323-t003:** Distribution of birds according to the lesion score among immunized groups, including their corresponding average lesion score and average body weight.

Group	No. of Birds		Lesion Scores				Average Lesion Score (±SE)	Average Weight (±SE)
		0	1	2	3	4		
P153	13	8	4	1	0	0	0.46 (0.18)	2066.77 (40.6)
P264	14	9	2	3	0	0	0.57 (0.23)	1952.50 (40.3)
P509	14	10	2	1	1	0	0.50 (0.25)	1946.79 (52.2)
P537	13	6	3	4	0	0	0.85 (0.25)	2087.69 (65.1)
P561	14	13	1	0	0	0	0.07 (0.07)	1900.71 (41.1)
Bacitracin	8	8	0	0	0	0	0.00 (0.00)	2790.00 (61.5)
Mix	12	10	0	1	0	1	0.50 (0.36)	2043.00 (37.9)
MLG_7820	9	2	3	4	0	0	1.22 (0.28)	2023.11 (54.9)
Quil-A	6	5	1	0	0	0	0.17 (0.17)	1925.00 (77.5)

**Table 4 animals-13-03323-t004:** Results of multiple comparisons between groups for lesion scores. Significant values (*p* < 0.05) are shown in bold and trends (0.05 ≤ *p* < 0.10) are presented as underlined.

	P153	P264	P509	P537	P561	Bacitracin	Mix	MLG_7820
P264	0.928	-	-	-	-	-	-	-
P509	0.802	0.784	-	-	-	-	-	-
P537	0.488	0.531	0.441	-	-	-	-	-
P561	0.213	0.216	0.273	0.082	-	-	-	-
Bacitracin	0.213	0.216	0.262	0.142	0.615	-	-	-
Mix	0.531	0.533	0.678	0.273	0.533	0.478	-	-
MLG_7820	0.176	0.220	0.184	0.493	**0.031**	0.060	0.150	-
Quil-A	0.493	0.493	0.622	0.262	0.641	0.492	0.943	0.142

**Table 5 animals-13-03323-t005:** Results of multiple comparisons between groups for body weight. *p*-values were corrected using the Benjamini-Hochberg method. Significant differences appear in bold characters (*p*-value < 0.05).

Compared Groups	Estimate	±SE	Z-Ratio	*p*-Value Adj.
P153—P264	114.33	65.90	1.734	0.226
P153—P509	120.05	65.90	1.821	0.208
P153—P537	−20.92	67.10	−0.312	0.855
P153—P561	166.12	65.90	2.520	0.062
P153—Bacitracine	**−723.11**	**77.20**	**−9.367**	**< 0.001**
P153—Mix	23.66	68.50	0.345	0.855
P153—MLG_7820	43.74	74.40	0.588	0.767
P153—Quil-A	142.20	85.00	1.672	0.235
P264—P509	5.71	64.70	0.088	0.965
P264—P537	−135.26	65.90	−2.052	0.148
P264—P561	51.79	64.70	0.801	0.632
P264—Bacitracine	**−837.44**	**76.20**	**−10.995**	**< 0.001**
P264—Mix	−90.67	67.40	−1.346	0.322
P264—MLG_7820	−70.59	73.30	−0.962	0.532
P264—Quil-A	27.87	84.10	0.331	0.855
P509—P537	−140.97	65.90	−2.138	0.140
P509—P561	46.07	64.70	0.713	0.692
P509—Bacitracine	**−843.16**	**76.20**	**−11.070**	**<0.001**
P509—Mix	−96.38	67.40	−1.431	0.296
P509—MLG_7820	−76.30	73.30	−1.040	0.487
P509—Quil-A	22.15	84.10	0.263	0.855
P537—P561	**187.04**	**65.90**	**2.837**	**0.028**
P537—Bacitracine	**−702.19**	**77.20**	**−9.096**	**< 0.001**
P537—Mix	44.59	68.50	0.651	0.729
P537—MLG_7820	64.67	74.40	0.869	0.591
P537—Quil-A	163.12	85.00	1.918	0.177
P561—Bacitracine	**−889.23**	**76.20**	**−11.675**	**< 0.001**
P561—Mix	−142.45	67.40	−2.114	0.140
P561—MLG_7820	−122.37	73.30	−1.668	0.235
P561—Quil-A	−23.92	84.10	−0.284	0.855
Bacitracine—Mix	**746.78**	**78.40**	**9.529**	**< 0.001**
Bacitracine—MLG_7820	**766.86**	**83.20**	**9.214**	**< 0.001**
Bacitracine—Quil-A	**865.31**	**92.70**	**9.335**	**< 0.001**
Mix—MLG_7820	20.08	75.60	0.266	0.855
Mix—Quil-A	118.54	86.10	1.377	0.315
MLG_7820—Quil-A	98.46	90.60	1.087	0.466

**Table 6 animals-13-03323-t006:** Results (*p* values) of multiple comparisons between groups for observed OTUs.

	Mix	MLG_7820	P153	P264	P509	P537	P561
MLG_7820	0.139	-	-	-	-	-	-
P153	0.087	0.554	-	-	-	-	-
P264	0.968	0.091	0.087	-	-	-	-
P509	0.459	0.087	0.087	0.539	-	-	-
P537	0.872	0.087	0.087	0.841	0.664	-	-
P561	0.968	0.377	0.211	0.872	0.554	0.904	-
Quil-A	0.872	0.377	0.459	0.872	0.293	0.554	0.852

**Table 7 animals-13-03323-t007:** Results (*p* values) of multiple comparisons between groups for Shannon index. Significant differences appear in bold characters (*p*-value < 0.05).

	Mix	MLG_7820	P153	P264	P509	P537	P561
MLG_7820	0.344	-	-	-	-	-	-
P153	0.566	0.608	-	-	-	-	-
P264	0.434	0.051	0.160	-	-	-	-
P509	**0.014**	**0.014**	**0.014**	0.066	-	-	-
P537	0.910	0.116	0.354	0.365	**0.014**	-	-
P561	0.921	0.160	0.442	0.365	**0.014**	0.836	-
Quil-A	0.365	0.051	0.116	0.836	0.354	0.365	0.365

**Table 8 animals-13-03323-t008:** Results of average alpha diversity indices (Observed, Shannon, and Inverse Simpson) for each experimental group (*p*-value < 0.05).

	Observed	Shannon	Inverse Simpson
P153	804	**3.67**	10.55
P264	721	3.59	10.1
P509	744	**3.7**	10.69
P537	706	**3.74**	11.63
P561	811	**3.93**	**13.32**
Mix	806	**3.86**	**13.2**
Quil-A	800	3.8	12.35
MLG_7820	828	**3.57**	**8.75**

**Table 9 animals-13-03323-t009:** Results (*p* values) of multiple comparisons between groups for inverse Simpson diversity index. Significant differences appear in bold characters (*p*-value < 0.05 and *p*-value < 0.01).

	Mix	MLG_7820	P153	P264	P509	P537	P561
MLG_7820	**0.0084**	-	-	-	-	-	-
P153	0.1129	0.4166	-	-	-	-	-
P264	0.0774	0.6288	0.6538	-	-	-	-
P509	0.2752	0.2741	0.7053	0.5526	-	-	-
P537	0.7053	0.0882	0.2752	0.1536	0.6288	-	-
P561	0.7222	**0.0302**	0.1137	0.0882	0.2127	0.5901	-
Quil-A	0.5428	0.3902	0.7053	0.6538	0.8201	0.8201	0.5526

**Table 10 animals-13-03323-t010:** Analysis of beta diversity on cecal microbiota using the ADONIS test based on the Bray-Curtis distance matrix. Significant differences between groups are highlighted in bold characters.

Comparison	*p*-Value
P509—P561	**0.004**
P509—Mix	**0.031**
P509—P153	**0.046**
P509—P264	**0.007**
P509—P537	**0.024**
P509—MLG_7820	0.077
P509—Quil-A	0.06
P561—Mix	0.064
P561—P153	**0.001**
P561—P264	**0.001**
P561—P537	**0.005**
P561—MLG_7820	**0.002**
P561—Quil-A	**0.017**
Mix—P153	**0.007**
Mix—P264	**0.006**
Mix—P537	**0.008**
Mix—MLG_7820	**0.046**
Mix—Quil-A	0.241
P153—P264	0.06
P153—P537	0.09
P153—MLG_7820	**0.019**
P153—Quil-A	**0.009**
P264—P537	**0.002**
P264—MLG_7820	0.053
P264—Quil-A	**0.015**
P537—MLG_7820	**0.009**
P537—Quil-A	**0.005**
MLG_7820—Quil-A	0.325

## Data Availability

Data is available in the SRA database as project ID: PRJNA734442.

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
