# Peer review of "Evaluation of the Immunoprotective Capacity of Five Vaccine Candidate Proteins against Avian Necrotic Enteritis and Impact on the Caecal Microbiota of Vaccinated Birds"

_animals, 2023, doi:10.3390/ani13213323_

Round 1

Reviewer 1 Report

This very valuable vaccine candidate study identified five recombinant proteins predicted to be surface-exposed and unique to necrotic enteritis-causing C. perfringens and the immunogenicity. A vaccine limiting the overgrowth of C. perfringens is an important step to improve broiler health in commercial production. If the birds were fed with a feed without anticoccidials, what was each their individual status? Were they, and if so, how severely, infected with Eimeria or were they even vaccinated? Why was Quil-A used and not a cytokine adjuvant or a TLR agonist? The discussion would benefit from broadening the picture towards the impact of Eimeria as one of the main predisposing factors for necrotic enteritis. Is competitive exclusion through a commercial probiotic additive after the vaccine a follow-up measure?

Minor comments: 

L 26 and 40 No abbreviations in the abstract 

l 47 zoonotic pathogen 

L 51 Please harmonise after introducing GIT as an abbreviation its use consistently in the manuscript. 

L 146 please move to the experimental section 

L 157 Why was the feed withdrawn?

L 166 Why was the inoculum dose different on the different days?

L 167 Please delete euthanised

Table 1 Please add the information given in L 174-L 184 to the table

L 224 Is the PCR protocol inhouse developed and was never published before? 

L 265 Why was an ANOVA not appropriate? 

L 343 See above. Why was the inoculum dose different on the different days?

Figure 3 Please harmonise the scale of P264 compared to P537, Why are there four mix groups if the figure caption states that 'Mix group received 50 μg of each of the four 420 candidate proteins (P153, P264, P509, P561, for a total of 200 μg of proteins) and 50 μg of 421 Quil-A adjuvant.'

Please shorten sentences to max. two lines (e.g. l 568-575, l 594-599 and elsewhere). 

Author Response

Dear Reviewer,

Thanks for your comments. Here are the answers:

  1. If the birds were fed a feed without anticoccidials, what was their individual status? Neither the Eimeria status was documented, nor the shedding of coccidia was not monitored. The experimental trial was conducted in level 2 research facilities that are submitted, between each trial, to rigorous cleaning and disinfection procedures targeting bacteria, viruses and parasites. We are confident that Eimeria cycling was either absent or extremely limited in the broiler chickens submitted to the experimental NE induction model and did not influence the results obtained.
  2. Were they, and if so, how severely, infected with Eimeria, or were they even vaccinated? Again, the shedding was not monitored, and birds were not vaccinated against Eimeria. Actually, Eimeria vaccination relies on the use of live oocysts that need to be managed properly to benefit from it; otherwise, this vaccination can have significant adverse effects on the intestinal health of broiler chickens, including predisposing them to necrotic enteritis. For these reasons, Eimeria vaccination was not used in the experimental design proposed in the current study.
  3. Why was Quil-A used and not a cytokine adjuvant or a TLR agonist? The Quil-A adjuvant was used since it enhances antibody response and activates T helper type 1 cells (Th1) and cytotoxic T lymphocyte (CTL)-mediated immune responses. In contrast, cytokine adjuvants and agonists of TLRs only activate some specific branches or mediators of the immune system.
  4. The discussion would benefit from broadening the picture towards the impact of Eimeria as one of the main predisposing factors for necrotic enteritis. Was modified accordingly. Please see the lines 565-569.
  5. Is competitive exclusion through a commercial probiotic additive after the vaccine a follow-up measure? Using probiotics in broilers has been shown to be beneficial for various aspects of birds’ health including improving zootechnical performances and protecting against pathogens through the modulation of the intestinal microbiota and immune system. As one of the aims of the current study was to evaluate the protective effect of the antibodies raised by the recombinant proteins tested against the adverse effects of virulent C. perfringens, no measure that could have acted as a confounding factor was used. The results of the current study indicated that immunization of broilers with the recombinant proteins tested had low impact on the intestinal microbial communities of vaccinated birds, with Lactobacillaceae even being increased in some experimental groups, relativizing the need for supporting this microbiota with a probiotic.

Minor comments: 

  • L 26 and 40 No abbreviations in the abstract: Modified accordingly. Please see lines 26 and 40. 
  • L 47 zoonotic pathogen: Modified accordingly. Please see line 49.
  • L 51 Please harmonise after introducing GIT as an abbreviation its use consistently in the manuscript. Modified accordingly. Please see lines and 673 and 693.
  • L 146 please move to the experimental section: The sentence has been deleted as it has already been mentioned in the experimental section (line 168).
  • L 157 Why was the feed withdrawn? Two main reasons for withdrawing the feed: i) the feed is withdrawn for 12 hours in order to create a stress on the birds which is recognized to increase their susceptibility to necrotic enteritis, and ii) the feed was withdrawn to replace it with a high-protein diet which is also recognized as another predisposing factor to necrotic enteritis).
  • L 166 Why was the inoculum dose different on the different days? Because the growth and consequently, the obtained bacterial concentration was different on every occasion the experiment was conducted. Overall, the different concentrations used in the current study were in the range of C. perfringens concentrations usually used to inoculate broiler chickens using a necrotic enteritis experimental induction model.
  • L 167 Please delete euthanized. Modified accordingly. Please see line 170.
  • Table 1 Please add the information given in L 174-L 184 to the table: Modified accordingly. Please see Table 1.
  • L 224 Is the PCR protocol inhouse developed and was never published before? It was already used and published in “Impacts of Short-Term Antibiotic Withdrawal and Long-Term Judicious Antibiotic Use on Resistance Gene Abundance and Cecal Microbiota Composition on Commercial Broiler Chicken Farms in Québec”. The reference was also added in lines 240 and 249.
  • L 265 Why was an ANOVA not appropriate? To test the effect of the group on the mass of individuals, a linear mixed model (LMM) was used including the identity of individuals as a random factor, in order to avoid potential pseudo-replication bias. Then the variance explained by the independent variable (the group) was analyzed via an ANOVA (using a Likelihood Ratio test here). Since the group had a statistically significant effect on mass (p < 0.001) and it was a qualitative variable with more than two modalities, a post-hoc test of multiple comparisons between the groups was then carried out (with a Benjamini-Hochberg correction on the P-values).
  • L 343 See above. Why was the inoculum dose different on the different days? Because the growth and consequently, the obtained bacterial concentration was different every day. But overall, the different concentrations used were in the range of C. perfringens concentrations usually used to inoculate broiler chickens using a necrotic enteritis experimental induction model.
  • Figure 3 Please harmonise the scale of P264 compared to P537. Modified accordingly. Please see Figure 3

  • Why are there four mix groups if the figure caption states that 'Mix group received 50 μg of each of the four 420 candidate proteins (P153, P264, P509, P561, for a total of 200 μg of proteins) and 50 μg of 421 Quil-A adjuvant.'? Because the obtained serum from the Mix group were tested against each candidate separately (e.g., one time against P153 antigens, the second time against P264, and so on).
  • Please shorten sentences to max. two lines (e.g., l 568-575, l 594-599 and elsewhere). Modified accordingly. Please see lines 593 and 626.

Reviewer 2 Report

The authors developed previously protein vaccine candidates for use against Clostridium perfringens (NE). In the current study, the vaccine candidates were tested against a NE challenge and both the microbiota and immune response were measured.

As with many studies concerning NE, challenged birds exhibit a wide range of responses to the challenge ranging from severe lesions and death, to what would appear to be no response to the challenge.

The gut microbiota data point to current lines of reasoning as to why certain birds/groups of birds show deleterious effects of NE and others do not, specifically concerning the buyrate- producing groups and the Enterbacteriaceae. 

Given this, the results are promising and point to a possible strategy to decrease NE problems in the poultry industry.

The science is sound, well written, and analysis, presentation and conclusions are appropriate.

The reviewer's only comment/question is why the authors did not measure GI IgA levels? 

Author Response

  • Why did the authors not measure GI IgA levels? Because of limited funds, the IgY response was prioritized. Our group has however ongoing research work that aims at better characterizing the immune response of broilers immunized with these candidate proteins.